ecology, environmental science

future landscape, ecosystem management, complex system, individual-based modelling, home-range, remote sensing

**Author for correspondence:**
Eduardo M. Arraut
e-mail: emarraut@ita.br

# Anticipation of common buzzard population patterns in the changing UK landscape

Eduardo M. Arraut[1,2], Sean W. Walls[3], David W. Macdonald[2] and Robert E. Kenward[4]

[1]Department of Hydric Resources and Environment, Civil Engineering Division, Aeronautics Institute of Technology, Praça Marechal Eduardo Gomes 50, 12228-900 São José dos Campos, SP, Brazil
[2]Wildlife Conservation Research Unit, Zoology Department, Oxford University, The Recanati-Kaplan Centre, Tubney House, Abingdon Road, Tubney, Abingdon OX13 5QL, UK
[3]Lotek-UK, The Old Courts, Worgret Road, Wareham, Dorset BH20 4PL, UK
[4]United Kingdom Centre for Ecology and Hydrology, Maclean Building, Benson Lane, Crowmarsh Gifford, Wallingford, Oxfordshire OX10 8BB, UK

EMA, 0000-0001-5323-4431; SWW, 0000-0002-8410-4784; DWM, 0000-0003-0607-9373; REK, 0000-0002-2769-1345

Harmonious coexistence between humans, other animals and ecosystem services they support is a complex issue, typically impacted by landscape change, which affects animal distribution and abundance. In the last 30 years, afforestation on grasslands across Great Britain has been increasing, motivated by socio-economic reasons and climate change mitigation. Beyond expected benefits, an obvious question is what are the consequences for wider biodiversity of this scale of landscape change. Here, we explore the impact of such change on the expanding population of common buzzards *Buteo buteo*, a raptor with a history of human-induced setbacks. Using Resource-Area-Dependence Analysis (RADA), with which we estimated individuals' resource needs using 10-day radio-tracking sessions and the 1990s Land Cover Map of GB, and agent-based modelling, we predict that buzzards in our study area in lowland UK had fully recovered (to 2.2 ind km$^{-2}$) by 1995. We also anticipate that the conversion of 30%, 60% and 90% of economically viable meadow into woodland would reduce buzzard abundance nonlinearly by 15%, 38% and 74%, respectively. The same approach used here could allow for cost-effective anticipation of other animals' population patterns in changing landscapes, thus helping to harmonize economy, landscape change and biodiversity.

## 1. Introduction

Worldwide, damaging imbalances in animal populations have been driven by anthropogenic landscape alterations that are now being further destabilized by climate change [1]. The impacts of such imbalances on ecosystem services and human livelihoods have already been severe, and will probably get more so. Examples span increases in crop pests, reductions in pollinators and extinctions of iconic species [1]. Hopes of remediation depend on advances in the understanding of restoration of wildlife, landscapes and their services [1,2]. Considering the connectedness of ecological processes, there are opportunities for powerful synergies. For example, reintroduction of a predator to mediate herbivore impacts on plants, or creation of urban green mosaics with culturally important animals in mind, can promote a cascade of desirable outcomes including climate change adaptation and mitigation, food production, water security, economic prosperity and human mental health [1,3,4]. Even a landscape modification as simple as planting trees in an agricultural area for predatory birds to roost can lead to pest control and thereby increase yield [5–7].

One example of land use change that is both important in itself and an insightful model from which wider lessons can be drawn is the increase by 5236 km$^2$, over the 25 years from 1990 to 2015, of woodland in the UK. This increase has come largely at the expense of grassland [8], motivated by woodland's role in flood management, timber's value for construction and vegetarianism/veganism reducing the need for grassland to feed livestock [9–11]. Recently, this afforestation process gained further momentum due to the imperative to mitigate climate change and associated pledges to plant millions of trees in the UK for carbon sequestration [12]. However, beyond expected socio-economic and climate benefits, an obvious question is what are the consequences for wider biodiversity of this scale of change in the UK landscape [13,14]. As ecosystems are complex, societies should consider holistically the cascading effects of their interventions.

The common buzzard *Buteo buteo* is a medium-sized generalist raptor that in the UK has suffered periodic declines due to human intervention. Over 500 years ago, they were regarded as vermin and payments were made for their corpses. Then gamebird rearing and shooting at the end of the eighteenth century eradicated buzzards from much of eastern Britain. This decline was further exacerbated in the 1950s by the depletion of a key resource, rabbits *Oryctolagus cuniculus*, infected with the myxomatosis virus [15]—this last impact was an unintended outcome of human actions which reached buzzards by cascading through the ecosystem. Nowadays, general attitudes towards buzzards in the UK are more positive, with only occasional, local episodes of human–wildlife conflict attributed to 'problem individuals', which prey on poultry or game birds [16]. The buzzard's resource requirements in lowland UK are clear cut: a tree for nesting or roosting, open areas of grassland, especially 'seasonally-long-grass' (meadow) and a combination of sparse grass, open shrub and dense shrub areas (rough-ground) for hunting [17]. The expansion of woodland under way in the UK is, thus, impacting buzzards in at least two opposing ways: an increase in potential roosting sites (trees), but a decrease in foraging sites (meadows). Our objective was to investigate how this change in key resources will play out for buzzards.

We built an agent-based model (ABM) in which virtual buzzards populated a land-cover map of the 1990s lowland UK. In this landscape, we also explored the consequences for the buzzard population of the gradual replacement of meadow fields of economically viable size (greater than or equal to 20 ha) by woodland, for timber extraction and climate change mitigation [11,18]. Model predictions were assessed by comparison with knowledge of wild buzzards according to five dimensions of their spatial ecology, namely home-range area, perimeter, pairwise overlaps and population distribution and abundance. Since two home-ranges can have the same areas and yet different perimeters, and vice versa, home-range area and perimeter are two proxies for energy expenditure from movement to acquire resources which could differ according to resource dispersion or buzzard foraging behaviour. Pairwise home-range overlaps are a proxy for territorial spacing. Our first results are predictions for buzzard maximum distribution and abundance in our focal study area in the 1990s, and in plausibly afforested future UK landscapes. In addition, we predict the geographical arrangements of buzzard territories and home-ranges. These individual-level insights have practical implications for, for example, the management of 'problem buzzards' via translocations. This opens the door to the general power of the novelty of our approach, which was

the creation of an ABM in which the resource parameters were estimated from wild buzzards using Resource–Area-Dependence Analysis (RADA). With RADA, we translated the Land Cover Map of Great Britain (LCMGB) of 1990 [19] into a map of buzzard key resources, estimated the minimum area of each key resource-containing category that the average individual buzzard needs, and discovered within which vicinity of the roosting site each food resource is typically found. The RADA process has conceptual parallels to the resource dispersion hypothesis, which postulates that territory size depends on the dispersion of the resources needed for survival and reproduction, and which has been used to explain the territories of more than 40 species in five continents [20]. Using the model presented here, predictions considering larger extents of lowland UK, or other realistic landscape change scenarios, could be explored to understand the impact of landscape change on individual buzzards and their populations. We think this same approach could be used efficiently to anticipate the consequences of landscape change on many other animals, thus aiding practical decision-making for biodiversity conservation and landscape management.

## 2. Material and methods

### (a) Brief characterization of buzzard space use

Individual buzzard home-ranges, resource use and reproduction were quantified in Dorset, southern UK. Between 1990 and 1995, 114 home-ranges were recorded from 72 radio-tagged buzzards. Nests were counted and transect (*T*) and mark-resighting (*M-R*) buzzard counts were conducted [15]. Each radio-tracked animal in this sample was wild, had not been previously trapped or relocated and was not preferably trapped over others on the basis of any characteristic other than our ability to reach the nest—they were therefore not STRANGE animals [21]. Home-ranges were estimated from standardized sets of 30 locations collected for each animal, by recording coordinates three times daily during a 10-day period that was either continuous or separated by a weekend. Estimations based on polygons, kernels and nearest neighbour (cluster) algorithms were tested. It was observed, for example, that home-ranges tend to be mononuclear [22]. Cores including more than 85% of locations seem to include excursions in flight not related to foraging, such as for social reasons [15]. In the study area, juveniles constituted about 10% of the population, and about 90% of them dispersed from the parental territory within 1 year of fledging, indicating their role in the partitioning of space and use of resources was minor when compared with adults [23]. After dispersal, buzzards try to establish their own territory, preferably in a previously unoccupied area [24]. If one is established, usually within the first 2 years of life, they will typically defend it throughout their lives [15]. No sex-based differences in resource use were found during the pre-breeding season [25]. The landscape at the time of tracking was a translation of the LCMGB of 1990 into a map depicting only the key resource-containing map categories for buzzards (electronic supplementary material, figure S2). This translation was for the resources identified by RADA [17].

### (b) Modelling

We followed a pattern-oriented modelling approach (POM) [26]. The model description follows the ODD (Overview, Design concepts and Details) protocol [27] and the model itself was created in NetLogo v. 6.0 [28] and analysed in R [29]. RADA was applied using Ranges 9 [30]. The electronic supplementary material is a TRACE document [31].

**Table 1.** Buzzard RADA-ABM entities, state variables, units and descriptions. When a state variable's composite name is separated by a dash, it defines a virtual buzzard characteristic or resource need, and when it is separated by an underscore it defines a resource map or the pseudo-number generator.

| entity | state variable | unit | description |
|---|---|---|---|
| pre-breeding buzzard | name | integer | ID |
| | my-roost | pixel | defended woodland pixel |
| | my-rgr | pixel | defended rough-ground pixels |
| | my-mead | pixel | defended meadow pixels |
| | $X$ | M | eastings of resource pixels forming territory |
| | $Y$ | M | northings of resource pixels forming territory |
| resource | map_cat | integer | map category ID of pixel in land-cover map |
| | resource | string | map category ID of pixel in buzzard resource map |
| | available? | Boolean | whether a resource pixel is free or defended |
| | searched? | Boolean | whether a resource pixel has had its free same-resource neighbouring pixels searched |
| global | rgr-area | integer | area of rough-ground required in inner range core (RADA) |
| | rgr-dist | M | constraint on distance from roosting site for rough-ground searches (but not for rough-ground defence) |
| | mead-area | integer | area of meadow required in outer core (RADA) |
| | mead-dist | M | constraint on distance from roosting site for meadow searches (but not for meadow defence) |
| | resource_data | pixel | LCMGB 1990 or land-cover change scenario to be translated into buzzard resource map |
| | m_1990 | pixel | buzzard resource map; translation of the LCMGB |
| | s_30% | pixel | scenario 1 (S1): 30% of economically viable meadow (>20 ha) turned into woodland |
| | s_60% | pixel | scenario 2 (S2): 60% of economically viable meadow (>20 ha) turned into woodland |
| | s_90% | pixel | scenario 3 (S3): 90% of economically viable meadow (>20 ha) turned into woodland |
| | map_eastings | M | E−W resource map extent |
| | map_northings | M | N−S resource map extent |
| | seed | integer | user-defined number (seed) for pseudorandom number generator |
| | seed_on | on/off | if 'on', pseudorandom number generator begins with user-defined seed, making run fully deterministic |

## (i) Purpose

Our primary purpose was to predict the distribution and abundance of common buzzards in the geographical space of the real landscape in lowland UK (i) in the 1990s, and (ii) in landscape change scenarios depicting a gradual replacement of meadow for woodland, motivated by socio-economic reasons and climate change mitigation. Our secondary purpose was to predict buzzard home-ranges and territories.

## (ii) Entities, state variables and scales

Entities and state variables are presented in table 1.

The study area was 22 km × 6 km (128 km$^2$ of land-cover and 4 km$^2$ of seawater) with grain size of 25 m × 25 m (the pixel size of LCMGB of 1990). Most of the land-covers occurring in lowland UK were present in the study area [25]. To avoid edge effects, a boundary strip of length equivalent to the mean largest span of wild buzzard home-ranges was included around the study area; virtual buzzards with settling-sites outside the study area could defend patches within it but were not considered in the abundance or distribution predictions, on the assumption that their incursive areas were balanced by excursive areas of those settled within the study area. Temporal scale was not explicitly represented because the focus was in discovering the maximum values for distribution and abundance of a saturated population, no matter how long it took for the buzzards to reach that density.

## (iii) Process overview and scheduling

The model's high-level algorithm is presented in figure 1.

1. The run starts with the unpopulated buzzard resource map, which is based on a translation of the LCMGB of 1990 using RADA.
2. A virtual buzzard settles on an undefended woodland patch, which becomes its roosting site. It will defend this patch (*my-roost* in table 1) and use it as the base from which to search for its key resources and establish a territory.
3. The virtual buzzard will then randomly choose a free rough-ground pixel within its rough-ground foraging distance (*rgr-dist*). In finding one, it will fly to it. This will be the seed from where it will search the entire landscape patch.
4. From the seed, the virtual buzzard begins an iterative search for its free neighbouring pixels of rough-ground, incorporating each into the territory (as *my-rgr*).
5. Then, when no free rough-ground neighbouring pixels are left, the virtual buzzard will search for rough-ground pixels at the edges of the raster patch it is incorporating into its territory and check whether these have free rough-ground neighbours. If any of them has, the virtual buzzard will iteratively move to each edge pixel with a free rough-ground neighbour, set it as a new base and apply the iterative neighbour search described in 4. When the landscape patch under scrutiny has been fully incorporated into the territory, the

**Figure 1.** High-level model algorithm.

virtual buzzard will move back to its roosting site and continue the search for additional free rough-ground patches (by repeating steps 3–5). This search will only stop when (i) the virtual buzzard has met its area requirements for rough-ground (*rgr-area*) and moved back to the roosting site, or (ii) it has not met these requirements and hence left the area.

6. The search for meadow begins from the roosting site and happens in exactly the same way as that for rough-ground (excepting, naturally, the changes to *mead-area* and to *mead-dist*). It stops when (i) the meadow resource needs (*mead-area*) have been met or (ii) they have not been met and hence the virtual buzzard left the area. When the meadow resource needs have been met, the territory will have been fully formed. Another virtual buzzard will then arrive at a random woodland patch which is as yet undefended and try to establish a territory by following the exact same rules (steps 2–6).

7. The exhaustion of free woodland patches, which causes the model to stop, leads to the emergence of maximum distribution and abundance in the landscape.

### (iv) Main model assumptions

The main assumptions of our model were that: (i) the random selection of the woodland patch for roosting does not lead to different maximum distribution and abundance than some other form of colonization; (ii) the wild buzzard does not actively avoid any land-cover that does not contain a key resource; (iii) the key resource-containing categories of LCMGB of 1990 were perfectly mapped (an assumption of RADA); (iv) there is a proportional relationship between patch area and amount of accessible resource (an assumption of RADA); (v) wild buzzards defend only the patches that provide them with the minimum amount of the needed resources.

### (v) Calibration

Parameter values, calibration intervals, steps and references are shown in table 2. Parametrization was restricted to the forage

**Table 2.** Model parameters (state variables), value and reference for value, and interval and step used in calibration. To establish a home-range in the study area, the virtual buzzard needed a tree to roost (woodland) and to search for enough rough-ground and meadow (area parameters) to meet requirements for small mammals and invertebrates.

| parameter | value | interval | step | source |
|---|---|---|---|---|
| my-roost | 0.06 ha | — | — | data (25 × 25 m pixel) |
| rgr-area | 0.56 ha | — | — | RADA |
| mead-area | 13.5 ha | — | — | RADA |
| rgr-dist | 500 m | 300–500 | 50 | data + calibration |
| mead-dist | 1200 m | 1150–1350 | 50 | data + calibration |

search distances for rough-ground and meadow. To capture home-range structure, we assessed the utilization distributions from 30% to 80% outlines at 5% intervals, because cores greater than 80% are associated with activities other than foraging, such as social interactions [22]. Initial values were based on a previous study [25] and final values were those which minimized the sum of the absolute differences between mean core% area of virtual and wild buzzards across the 11 cores. Tests with up to 50 model runs per parameter value combination showed 6 runs sufficed for obtaining reasonably stable results while considerably reducing runtimes.

### (vi) Landscape change scenario

Maps of landscape change scenarios represented the rural economy gradually shifting from meadow to woodland, motivated by, inter alia, woodland's role in flood management, timber's value for construction and vegetarianism/veganism reducing the need for grassland to feed livestock [9–11]. The smallest economically viable plot for meadow conversion to woodland was considered to be 20 ha [18]. Scenarios represented 30%, 60% and 90% conversion of randomly chosen meadow plots larger than 20 ha into woodland.

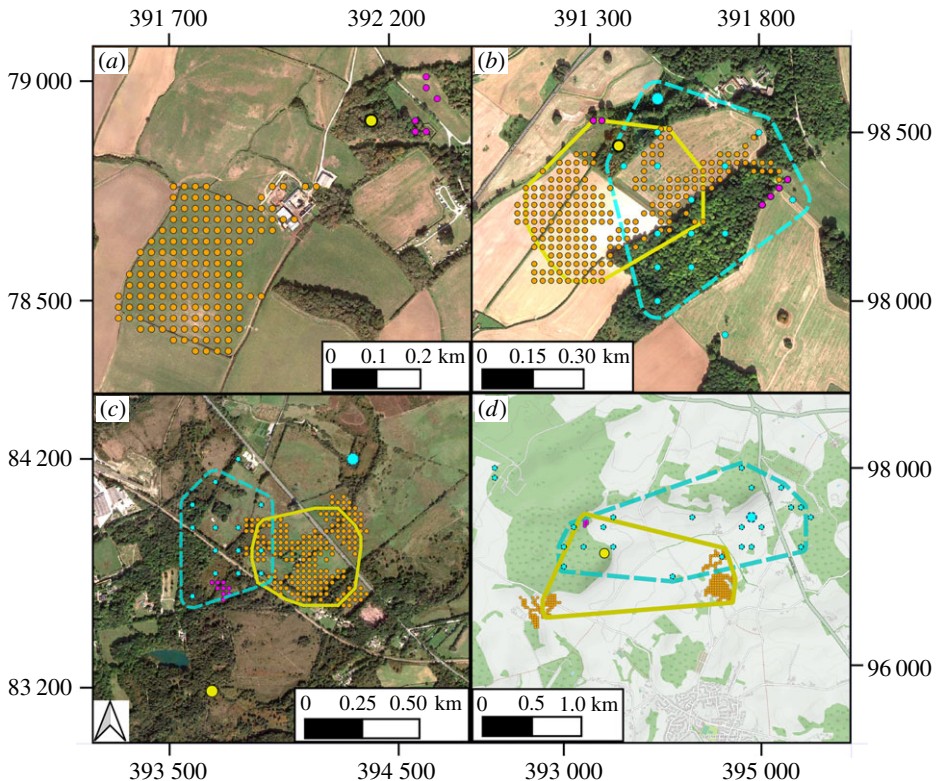

**Figure 2.** Virtual and wild buzzards in lowland UK. (*a*) Nuclear area of a virtual buzzard's home-range, formed by roosting site (yellow, pale) and nearby defended patches of rough-ground (purple, dark) and meadow (orange, mid-tone). In each of (*b–d*), one virtual and one wild buzzard (blue, dashed) occupy adjacent home-ranges of similar size and shape as defined by 80% convex polygons. Background maps in (*a–c*) are recent Google Satellite (less than 1 m) [35] and (*d*) is an Open Street Map [36] as often used to inform policy. Buzzards' data projected using 1936 British National Grid (EPSG 27700), with location resolution: virtual = 25 m; wild = 100 m. (Online version in colour.)

### (vii) Output verification

The model was assessed qualitatively and quantitatively with regard to producing virtual buzzards with home-ranges with size, shape and pattern of overlap similar to wild buzzards. Virtual and wild buzzard home-range cores and territories were plotted in a GIS to visually assess these three characteristics of home-range structure relevant to defence of resource patches. Virtual buzzard pairwise home-ranges overlaps, a proxy for their territorial behaviour and an emergent pattern in the model, were compared with those of wild buzzards using two-tailed tests [32].

### (viii) Sensitivity analysis

Local sensitivity analysis (LSA) and global sensitivity analysis (GSA) were performed. LSA used a modified version of the Morris screening method, which makes no assumptions about the model and uses individually randomized one-factor-at-a-time designs to assess the effects of changes in parameter values on outputs [33,34]. The modified Morris screening was used to assess the relative importance of rough-ground area and rough-ground forage search distance, and meadow area and meadow forage search distance, on each of six individual- and population-level model outputs: abundance, 80% home-range core overlap percentage, 40% home-range core area, 80% home-range core area, 40% home-range core perimeter, 80% home-range core perimeter (explanation of the modified Morris Screening method is presented in the TRACE document in electronic supplementary material). Parameter values for rough-ground forage search distance and area, and for meadow forage search distance and area, were varied around the reference values by, respectively, 70 and 67, 67 and 67%. GSA was based on a full factorial design and aimed at assessing possible interactions between the two main parameters influencing abundance, namely meadow area and forage search distance; meadow area: min = 9.5 ha, max = 17.5 ha, step = 1 ha, and meadow forage search distance: min = 800 m, max = 1600 m, step = 100 m.

### (ix) Output corroboration

Predictions for maximum abundance were compared with estimates obtained via mark-resighting and radio-corrected distance-transects carried out in 1995–1996 [23]. The final spatial distributions of home-ranges and territories were also visually compared to those of the sample of wild buzzards.

## 3. Results

### (a) Individual-level predictions: home-range structure

Home-ranges of virtual buzzards were similar to those of wild buzzards visually (figure 2) and quantitatively (figure 3). The relative frequency distributions for size and perimeter of 80% convex polygons for both virtual and wild buzzards were positively skewed, with close means, medians and inter-quartile ranges. As calibration results indicate, the virtual buzzards' inner core (40% polygon), which was associated most strongly with rough-ground, was also similar to that of wild buzzards (electronic supplementary material, figure S8). As with the wild birds, about 70–80% of virtual buzzards had compact resource-associated cores in the richer meadow zones. The rest of the wild and virtual animals occurred where rough-ground or meadow were thinly spread, for example, due to another buzzard's territory, an urban zone or a large patch of arable land being in the way. These formed 80% convex polygons one order of magnitude larger than the more compact ones, which is a significant variation in terms of possibilities for occupying space.

The collective territorial pattern of virtual buzzards, which was emergent in the model, was also similar to that of wild buzzards (figure 3; electronic supplementary material, figures S4–S6). Again, this was true in terms of

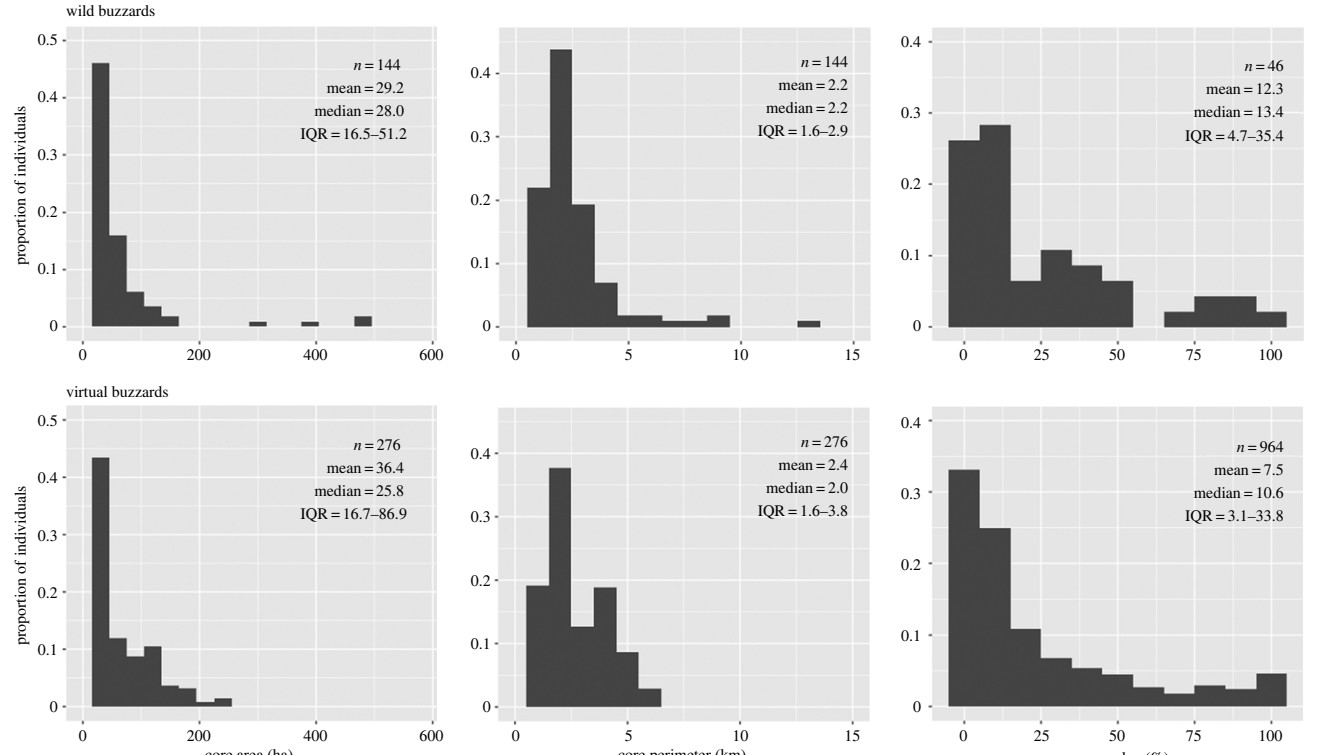

**Figure 3.** Comparison of wild and virtual buzzards' relative frequency distributions for home-range core area, perimeter, and pairwise overlaps of 80% range cores (as proxy for territorial spacing), with sample size (*n*), mean, median and inter-quartile range (IQR). To improve visibility, the largest home-range of a wild buzzard, which was 1270 ha, was omitted. Core area was calibrated based on mean values only, not the shape of the distribution, and neither perimeter nor overlap were subject to calibration (see Calibration in electronic supplementary material).

**Table 3.** Abundances of buzzards in the 1990s landscape (UK), with scenarios of (S1) 30%, (S2) 60% and (S3) 90% conversion of meadow into woodland (100 runs each). For comparison, field-based estimates using transect-counting (*T*) and mark-resighting (*M-R*) were obtained from surveys carried out during 1995–1996 [23].

|  | *T* | *M-R* | UK | S1 | S2 | S3 |
|---|---|---|---|---|---|---|
| abundance (mean) | 256 | 250 | 275 | 235 | 170 | 71 |
| 95% CI | 152–435 | 82–417 | 274–276 | 234–236 | 169–171 | 70–72 |
| range (min–max) | — | — | 264–287 | 222–245 | 158–181 | 63–77 |

positive skewness, means, medians and inter-quartile ranges, and overlap between neighbouring home-ranges (Mann–Whitney *U*-test, two-tailed, pairwise, involving overlaps between 114 wild and 276 virtual buzzards and null hypothesis that there was no difference between the two samples yielded: $W = 24\,548$, $p = 0.219$). Thus, in addition to roosting on a woodland patch and defending the same key resources, virtual buzzards also shared space with neighbours similar to wild buzzards.

### (b) Population-level predictions: abundance and distribution within the 1994 landscape

The mean maximum abundance over 100 runs was about 7% and 10% larger than the mean abundance based on transect-counting or mark-resighting, respectively (table 3; electronic supplementary material, figures S6 and S7). The distribution of virtual buzzards encompassed that of wild buzzards and extended to areas where they were not tracked (electronic supplementary material, figure S6).

### (c) Population-level predictions: abundance and distribution within landscape change scenarios

Predictions for 30%, 60% and 90% of meadow fields (of greater than 20 ha) being converted into woodland were that the number of buzzard territories within the study region would decline nonlinearly by 15%, 38% and 74%, respectively (table 3). The reason was primarily that, with reduction and dispersion of meadow, the smaller cores that once abounded became unviable (figure 4).

### (d) Sensitivity analysis

LSA showed the main parameter influencing abundance, and in an inverse way, was the individual's meadow area requirement (electronic supplementary material, figures S9 and S10). Rough-ground area also inversely influenced abundance, but much less strongly. Pairwise home-range core overlaps depended mainly and inversely on how far a buzzard could search for meadow, though overall variability was less than 7%. Home-range area and perimeter lengths of the

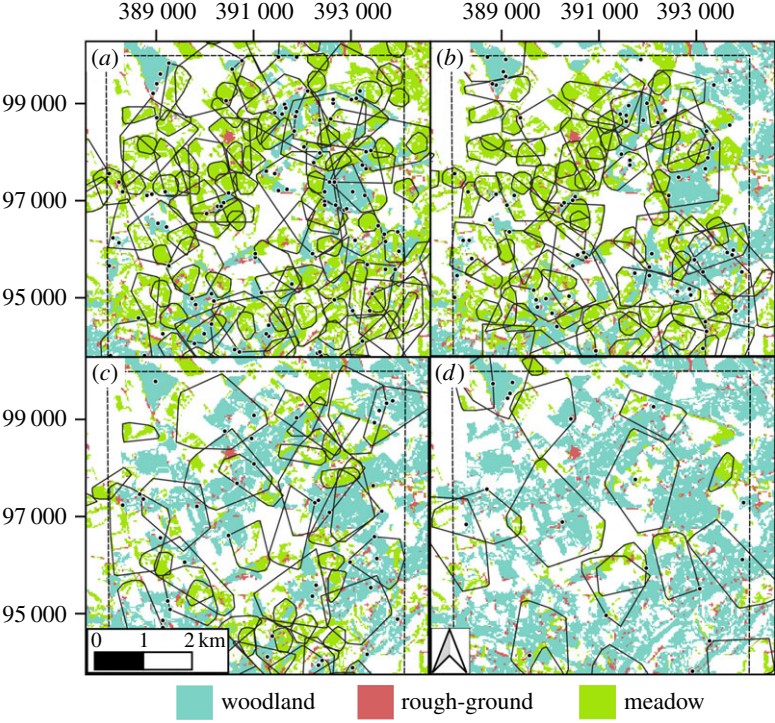

**Figure 4.** Partial view of predictions for common buzzard maximum abundances and distributions: (*a*) landscape at the time of tracking, and conversion of (*b*) 30%, (*c*) 60% and (*d*) 90% of meadows larger than 20 ha into woodland, for timber production. Note that roosts (dots) can lie outside core foraging ranges, but not outside the (dashed) study area. Data projected using 1936 British National Grid (EPSG 27700). (Online version in colour.)

inner and outer cores depended more strongly on how much meadow the individual required, and how far it would go to find it, the latter prevailing with regard to outermost core areas and perimeter lengths. We note that despite the small effect on outputs of rough-ground area and forage search distance, these parameters were kept because they refined results and made the model applicable to landscapes with greater variation in structure.

## 4. Discussion

In the study area, the mean virtual buzzard abundance (100 runs) was between 7% and 10% larger than field estimates based on Transect-counting and Mark-resighting, respectively (table 3). A few isolated places within the study area, corresponding to about 5% of its extent, were not easily accessed during fieldwork for radio-tagging or population censusing, and the model predicted they could accommodate buzzards. Discounting the number of buzzards predicted within these areas reduces the difference between model and field estimates to roughly 5%, which is within the uncertainty associated with each technique (table 3), and hence unlikely to be biologically meaningful. We thus conclude that by 1995, at least, buzzards in the area had recovered to maximum density.

When the landscape change scenarios were considered, buzzard abundance in the study area decreased steeply and nonlinearly with the increased conversion of meadow patches of economically viable size (greater than 20 ha) into woodland (table 3 and figure 4). As explained earlier, a report from UK Centre for Ecology and Hydrology (UKCEH) showed that over the last 25 years, land uses in Great Britain and Northern Ireland have changed from arable (−0.3%) and, mainly, grassland (−3.0%) to urban (+1.3%) and woodland (+2.0%) [8]. Therefore, in areas where buzzard numbers have built to

maximum capacity and meadow is the limiting resource, a loss of meadow is expected to have led to a decline in density. However, this decline is expected to have happened in only a few places as yet, because in most areas of lowland UK buzzard densities are still increasing.

### (a) Buzzard home-range structuring and territoriality

The question of what influences a buzzard's territory in lowland UK has been considered for over 40 years, with hypotheses about the roles of resource accessibility, social excursions and territorial disputes [37]. The results here further corroborate our earlier analysis [15], which indicated that the territory consists of a patch of woodland for roosting and enough rough-ground and meadow to meet minimum requirements for hunting small mammals and invertebrates (figure 2). This fits with the interpretation that common buzzards follow a contractionist [38] or area-minimizing home-range strategy [39], where ranges are tightly shaped by resource dispersion. Other examples of animals with a similar home-range strategy are goshawks *Accipiter gentilis* [40], Blandford foxes *Vulpes cana* [41], some populations of spotted hyaenas *Crocuta crocuta* [38], African lions *Panthera leo* [42] and female black bears *Ursus americanus* [43]. Naturally, animals following this strategy are particularly affected by changes to landscape structure, as exemplified here (figure 4).

### (b) Model uncertainties and possible improvements

For distribution and abundance, uncertainty in prediction was associated mainly with (i) the estimation of meadow area requirement and (ii) how well the virtual buzzard's territorial behaviour represented that of the wild buzzard. The precision of the meadow area requirement depended on the applicability of the assumptions that the LCMGB of 1990 was fully accurate and of there being a proportional

relationship between patch area and resource accessibility, both of which depended to a large extent on the quality of home-range estimation and mapping [19,22]. In practice, omission error in the mapping of meadow would lead to the underestimation of the minimum individual requirement and hence to the overestimation of abundance, while commission error would have the opposite effect. The representation of territories, in its turn, was based on hypothesis testing using variants of the ABM that differed by how the virtual buzzard defended its key resources (electronic supplementary material, figures S4 and S5). In addition, calibration of forage search distances may have led to overfitting that could be minimized, and the model made more general, with the use of energy budgets [44,45].

Predictions for the change scenarios did not consider possible functional relationships with other species [46]. For example, large proportional increases in woodland area may lead to more (re-introduced) goshawks, which compete with buzzards for woodland and edge prey and, being also a predator of buzzards, may deter buzzards from foraging for worms in meadow near woodland [47]. Such a further reduction in buzzard density would stem from a mechanism not included in our model.

### (i) Anticipation of animal population patterns in changing landscapes

Resource selection functions (RSFs) seem to be the most popular method to predict animal distribution and abundance in the geographical space of a real landscape. An RSF can predict these population patterns from correlations between a wide variety of data about animal presence (e.g. spoor or GPS tracking) and for resources or conditions (e.g. land-cover or altitude) [48,49]. Important assumptions of an RSF are that (1) the animal population is at equilibrium density or following an ideal free distribution when the calibration data are collected, (2) abundance does not depend on factors other than resources, and (3) the availabilities of the resources in the calibration and extrapolation landscapes are similar [48]. An example of when assumptions (1 and 2) would not have been reasonable was shown with an ABM for oystercatchers in the Exe estuary, UK. Mortality was found to be influenced by interference competition only after a certain density threshold, so an RSF built using data collected when density was below this threshold would overestimate maximum abundance by a considerable margin [45]. Additionally, when the resources are found within a landscape category that is being impacted by human action, e.g. meadow for buzzards being replaced by woodland, assumption (3) may be hard to meet and, therefore, extrapolations to future landscapes may be problematic. Thus, the assumptions underlying an RSF can restrict applicability to certain situations that may be particularly important for conservation, such as when an animal population is below equilibrium density owing to endangerment or recurrent perturbations, or when the aim is to assess the impact of landscape change on wildlife.

ABM offers greater flexibility by allowing for the explicit representation of an ecological mechanism that connects animal space use with landscape structure—the procurement by individual organisms of the resources needed to survive and breed [26,44,50,51]. However, carrying out the fieldwork required to identify and quantify individual animals' resource needs has often been challenging. Thus, in pioneering landscape-explicit ABM resource-based parameters were assumed, as for red and grey squirrels *Sciurus vulgaris* and *S. carolinensis* [52], or four small mammal species in the UK [53]. Alternatively, resource-based parameters were supported by field-data collected over decades, e.g. oystercatchers *Haematopus ostralegus*, UK [54], skylarks *Alauda arvensis* in Denmark [55] and river salmonids in California, USA [56], sometimes across thousands of kilometres squared, e.g. grey wolves *Canis lupus* in the Italian alps [57], African elephants *Loxodonta africana* in the Kenya–Tanzania border [58] and tigers *Panthera tigris* in Nepal's Chitwan National Park [59].

The strength of the ABM approach we applied to buzzards comes from having estimated the virtual animals' resource needs from wild animals using RADA, which relies on remote animal tracking and mapping of the resources. Such use of remote sensing to assess individuals' resource requirements has the potential to allow for more efficient estimation of the ABM parameters, and to considerably increase the range of species that can be simulated [60–62]. For example, the rapid recent reduction in the size of GPS tags has been revealing the intricate home-ranges of small birds, such as of the European nightjar *Caprimulgus europaeus* that weighs around 60 g [63]. In turn, populations of a single tree species have been mapped on the basis of variation in crown shape and phenology with very high-resolution (less than 1 m) satellite imagery [64,65], while land-cover mapping with spatial resolution suitable for detailed analysis of animal resource use (10–30 m) is increasingly available for many countries, continents and even the entire planet [8,66–70]. Importantly, RADA has worked with small sample sizes, as shown by results for data gained in single years during 10-day tracking sessions also for 15 red and 17 grey squirrels, and for contractionists or area-minimizing home-range strategists, it is expected to work even when the population is below equilibrium density [17]. Additionally, citizen-scientists can get involved in tracking or mapping [71], thus facilitating bottom-up management of landscapes and species, which across 34 mainly European local studies was found to have more influence than top-down management on the sustainability of local biodiversity and ecosystem services [72]. Indeed, IUCN nowadays recommends planning 'nature-based solutions' at the landscape scale and considering local knowledge [73].

Our results have shown that replacing meadow with trees is likely to reduce the buzzard density in the area we studied. We can apply the model to other areas of the UK if land managers want to consider the wider effects of planting trees on a prominent and protected species, or develop similar models of other species tracked in appropriately mapped areas. Anticipating that a crucial population will prosper, or an agricultural pest collapse, can help harmonize economic development with animal conservation and ecosystem service provision. We thus hope that the breadth, efficiency and simplicity of the modelling approach used here may contribute to addressing the more general conservation paradox that 'we are not limited by lack of knowledge but failure to synthesize and distribute what we know' [74].

**Data accessibility.** The following data files are available for public use through the Dryad Digital Repository: https://doi.org/10.5061/dryad.8n183 [75]. ALBAUT (with any extension, location data from 114 radio-tagged buzzards). Land Cover Map of Great Britain of 1990 is available against permission at public site, https://www.ceh.ac.uk/services/land-cover-map-1990.

**Authors' contributions.** E.M.A.: conceptualization, formal analysis, funding acquisition, investigation, methodology, resources, software,

validation, visualization, writing—original draft, writing—review and editing; S.W.W.: data curation, investigation, validation, writing—original draft, writing—review and editing; D.W.M.: funding acquisition, investigation, project administration, resources, supervision, writing—original draft, writing—review and editing; R.E.K.: conceptualization, data curation, funding acquisition, investigation, methodology, resources, supervision, validation, visualization, writing—original draft, writing—review and editing.

All authors gave final approval for publication and agreed to be held accountable for the work performed therein.

Competing interests. We declare we have no competing interests.

Funding. Funded by National Institute of Science and Technology for Climate Change (INCT-MC), Brazilian Network on Global Climate Change Research (Rede CLIMA), Brazilian National Research Council (CNPq), UK Natural Environment Research Council and WildCRU (Oxford University).

Acknowledgements. We appreciated comments from Jose Luis Arraut, Paul Johnson, Nick Casey, Seth Tisue, James Steiner, Jim Lyons, Luiz Aragão, Ana Carolina Garcez de Castro, the associate editor and two anonymous referees. Map data copyrighted Thunderforest and OpenStreetMap contributors and available from https://www.openstreetmap.org.

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
