## [Peer Review File · Proceedings of the Royal Society B: Biological Sciences]

Review History

RSPB-2020-2748.R0 (Original submission)

Review form: Reviewer 1

Recommendation

Major revision is needed (please make suggestions in comments)

Scientific importance: Is the manuscript an original and important contribution to its field?

Good

General interest: Is the paper of sufficient general interest?

Acceptable

Quality of the paper: Is the overall quality of the paper suitable?

Marginal

Is the length of the paper justified?

Yes

Should the paper be seen by a specialist statistical reviewer?

Yes

Do you have any concerns about statistical analyses in this paper? If so, please specify them explicitly in your report.

No

It is a condition of publication that authors make their supporting data, code and materials available - either as supplementary material or hosted in an external repository. Please rate, if applicable, the supporting data on the following criteria.

Is it accessible?

Yes

Is it clear?

No

Is it adequate?

No

Do you have any ethical concerns with this paper?

No

Comments to the Author

The authors use buzzard tracking data in an agent-based model to predict the distribution and estimate the potential population of buzzards in a 22 km x 6 km area of southern Dorset. They also project the potential changes in population if areas of 30%, 60% and 90% of meadow are converted to woodland.

The model developed here seems to be properly formulated and implemented. The comparison with empirical data is quite good. Predicted abundance is 9% greater than field-based estimates. Such a small difference could easily be explained by a number of factors, so it and other comparisons seem to validate the model. Figure 4 is very nice.

One problem that I have with the manuscript is that it seems to be a 'methods paper' rather than addressing an ecological issue. It is true that the results seem to confirm hypotheses on the habitat needs of the buzzards. But the manuscript seems mostly aimed at showing how well the methodology works. The 30%, 60% and 90% hypothetical conversions of meadow to woodland appear to be projected merely for providing something to apply the model to. I think that a lot more could be learned from this exercise by using the results so far, or some extensions, to explore aspects of landscape ecology theory, such as a more thorough study of how various landscape factors, including configuration of habitat types, affect consumer abundance.

A second problem is that the manuscript is very poorly written including a failure to clearly define some of main parameters and assumptions.

I have detailed comments below. Major and minor comments are mixed. Page numbers used in this review refer to the numbers at the bottom of the page.

Page 3

Abstract, Line 6. I think change 'when animal and fieldsite were small' to 'when animals and fieldsites were small', although I am not sure.

Page 5

Line 11. 'three degrees of realisation'. I am not sure what those are as I didn't see reference to that term later. I am guessing they are size, shape and pattern of overlap, but I am not sure.

Line 20. Delete comma after 'UK'

Page 6.

Somewhere around here, or perhaps earlier, the authors need to provide a paragraph or so on buzzard ecology; at least enough to have a succinct explanation of how they use habitat types. As it is, the reader has to piece together bits of information scattered through the manuscript.

Page 7

Lines 10-11 "in which meadow is gradually substituted for wood". I think the opposite is meant, "in which wood is gradually substituted for meadow".

Line 16. 'rough ground'. I could find no place where that is defined.

Page 9

Line 4. Change 'from where' to 'from which'

Page 10

Lines 16-18 "we assessed the utilization distributions as outlines at 5% intervals, from 30% to 80%, using 80% as proxy for the smallest core encompassing all resources because buzzard behaviour involves more flight activity in the outermost 15% of locations." I don't understand this sentence. It needs to be revised.

Page 12.

"to assess the relative importance of rough-ground area and distance and meadow area and distance". I am not sure what is meant by 'distance' here. The word 'distance' is used in various ways in the manuscript and it is often not clear what it means in particular cases.

Page 14

Line 2. Insert comma after 'neighbors'

Line 17. 'spreading out of meadows'. Does that mean that the density of patches of meadows is decreasing?

Page 15

Lines 2-3. "LSA showed that abundance was inversely regulated mainly by the individual's meadow area requirement." I am not sure what the term 'inversely regulated' means and I don't really understand Figures S9 and S10. I don't know what Morris Screening is in Figure S9 - it should be explained. In Figure S10, which I magnified to the maximum possible on my laptop, but could still only read 'MCP' on the y-axis, and I don't know what that means - some sort of perimeter perhaps. The figure captions here are inadequate.

Lines 15-17. "Modelling alone seems insufficient, because in applied cases not only the qualitative population patterns, but also the quantitative one pertaining to the animal and landscape in question, are important." The last part of the sentence seems like a non sequitur, as modelling provides quantitative patterns.

Page 17

Line 9. Change 'uncertainty in predictions were' to 'uncertainty in prediction was'

Page 18

Line 2. I am not sure what 'worming' means here. Do buzzards eat worms?

Therefore, on the one hand I think that some of the model output that I could understand is very nice. On the other hand, the study comes off as a 'methods' paper and is poorly written. At the very least the manuscript needs a thorough revision. Maybe there is more that the authors could do with the results to link the results to some more general landscape level principles. Otherwise, I would recommend that the authors, after thorough revision, submit the manuscript to a journal like "Methods in Ecology and Evolution" or "Ecological Modelling"

Review form: Reviewer 2

Recommendation

Accept with minor revision (please list in comments)

Scientific importance: Is the manuscript an original and important contribution to its field?

Good

General interest: Is the paper of sufficient general interest?

Excellent

Quality of the paper: Is the overall quality of the paper suitable?

Good

Is the length of the paper justified?

Yes

Should the paper be seen by a specialist statistical reviewer?

No

Do you have any concerns about statistical analyses in this paper? If so, please specify them explicitly in your report.

No

It is a condition of publication that authors make their supporting data, code and materials available - either as supplementary material or hosted in an external repository. Please rate, if applicable, the supporting data on the following criteria.

Is it accessible?

Yes

Is it clear?

Yes

Is it adequate?

Yes

Do you have any ethical concerns with this paper?

No

Comments to the Author

Anticipation of animal population patterns in a changing landscape

Eduardo M. Arraut, Sean W. Walls, David W. Macdonald and Robert E. Kenward

The paper illustrates well how data obtained with Resource-Area-Dependence Analysis (RADA), which quantifies the resource needs of individuals, can be used in Agent-based Modelling (ABM) to anticipate changes in animal populations in response to landscape alterations. The paper also illustrates how this approach can be used to predict how animal populations are likely to respond to various scenarios of landscape change.

The model species in the buzzard, a bird for which the authors have good data on the use of landscapes. The paper is generally well written, and the figures are adequate. However, I believe that there are a few issues that should be addressed to improve the paper.

The introduction is formatted somewhat like a “state of the art” and some key elements usually included in the introductions of papers are missing. Most obvious is the lack of the presentation of the objectives of the paper. These are apparently included only in the modelling subsection of the Methods, but readers usually look for them in the introduction. I also believe that it would be important to add to the introduction a paragraph explaining what is new in the paper. Why is it important to carry out this study? How does it differ from other modeling studies and approaches?

The discussion section includes a good text about the limitations of the modelling approach used in the study. However, I believe that it would gain by including a discussion on the relative merits of the proposed approach compared to other existing modelling approaches with similar objectives. Based on the results obtained in the study, can you recommend the approach followed? Are the results likely to be better than those that you could have obtained using other existing approaches? Why? Under what circumstances is the proposed one better?

A few specific comments:

P5L8 “In a previous study, we used remote sensing data collected over five years to apply Resource-Area-Dependency Analysis (RADA), using the buzzard tracking data and the Land Cover Map of Great Britain of 1990s.”

Comment: Remote sensing is nowadays a very broad concept... I suggest that you indicate what type of remote sensing data you are referring to and what it was used for, in the modelling.

P8L6 Our general purpose was to test whether an ABM of a wide-ranging animal...

Comment: I suggest that you avoid using the word “test”, because strictly speaking you did not really test this.

P12 L13 “2. Entities, state variables and scales”

Comment: I believe that most of the information in this section would be easier to consult if it were presented as a table or diagram.

Decision letter (RSPB-2020-2748.R0)

16-Dec-2020

Dear Dr Moraes Arraut:

I am writing to inform you that your manuscript RSPB-2020-2748 entitled "Anticipation of animal population patterns in a changing landscape" has, in its current form, been rejected for publication in Proceedings B.

This action has been taken on the advice of referees, who have recommended that substantial revisions are necessary. With this in mind we would be happy to consider a resubmission, provided the comments of the referees are fully addressed. However please note that this is not a provisional acceptance.

Sincerely,
 Dr Robert Barton
 mailto: proceedingsb@royalsociety.org

Associate Editor
 Board Member: 1
 Comments to Author:

I looked at the ms and the reviews. I like about this ms is the integration of data and modelling and making it work. This is interesting and important. However, the ms would be much more far reaching if the results address wider ecological issues or questions. The reviewers make the same points. Addressing these points will require changes to introduction and discussion. At the moment, the paper has an interesting method but as it is I do not think that is of sufficient interest for the readership of this journal. To make it suitable for a wide biological audience the focus should come away from the just the method and include wider ecological aspects. The paper would benefit if additional hypotheses would be tested, or if it could explore wider landscape ecology aspects.

Reviewer(s)' Comments to Author:
 Referee: 1

Comments to the Author(s)

The authors use buzzard tracking data in an agent-based model to predict the distribution and estimate the potential population of buzzards in a 22 km x 6 km area of southern Dorset. They also project the potential changes in population if areas of 30%, 60% and 90% of meadow are converted to woodland.

The model developed here seems to be properly formulated and implemented. The comparison with empirical data is quite good. Predicted abundance is 9% greater than field-based estimates. Such a small difference could easily be explained by a number of factors, so it and other comparisons seem to validate the model. Figure 4 is very nice.

One problem that I have with the manuscript is that it seems to be a 'methods paper' rather than addressing an ecological issue. It is true that the results seem to confirm hypotheses on the habitat needs of the buzzards. But the manuscript seems mostly aimed at showing how well the methodology works. The 30%, 60% and 90% hypothetical conversions of meadow to woodland appear to be projected merely for providing something to apply the model to. I think that a lot more could be learned from this exercise by using the results so far, or some extensions, to explore aspects of landscape ecology theory, such as a more thorough study of how various landscape factors, including configuration of habitat types, affect consumer abundance.

A second problem is that the manuscript is very poorly written including a failure to clearly define some of main parameters and assumptions.

I have detailed comments below. Major and minor comments are mixed. Page numbers used in this review refer to the numbers at the bottom of the page.

Page 3

Abstract, Line 6. I think change 'when animal and fieldsite were small' to 'when animals and fieldsites were small', although I am not sure.

Page 5

Line 11. 'three degrees of realisation'. I am not sure what those are as I didn't see reference to that term later. I am guessing they are size, shape and pattern of overlap, but I am not sure.

Line 20. Delete comma after 'UK'

Page 6.

Somewhere around here, or perhaps earlier, the authors need to provide a paragraph or so on buzzard ecology; at least enough to have a succinct explanation of how they use habitat types. As it is, the reader has to piece together bits of information scattered through the manuscript.

Page 7

Lines 10-11 "in which meadow is gradually substituted for wood". I think the opposite is meant, "in which wood is gradually substituted for meadow".

Line 16. 'rough ground'. I could find no place where that is defined.

Page 9

Line 4. Change 'from where' to 'from which'

Page 10

Lines 16-18 "we assessed the utilization distributions as outlines at 5% intervals, from 30% to 80%, using 80% as proxy for the smallest core encompassing all resources because buzzard behaviour involves more flight activity in the outermost 15% of locations." I don't understand this sentence. It needs to be revised.

Page 12.

"to assess the relative importance of rough-ground area and distance and meadow area and distance". I am not sure what is meant by 'distance' here. The word 'distance' is used in various ways in the manuscript and it is often not clear what it means in particular cases.

Page 14

Line 2. Insert comma after 'neighbors'

Line 17. 'spreading out of meadows'. Does that mean that the density of patches of meadows is decreasing?

Page 15

Lines 2-3. "LSA showed that abundance was inversely regulated mainly by the individual's meadow area requirement." I am not sure what the term 'inversely regulated' means and I don't really understand Figures S9 and S10. I don't know what Morris Screening is in Figure S9 - it should be explained. In Figure S10, which I magnified to the maximum possible on my laptop, but could still only read 'MCP' on the y-axis, and I don't know what that means - some sort of perimeter perhaps. The figure captions here are inadequate.

Lines 15-17. "Modelling alone seems insufficient, because in applied cases not only the qualitative population patterns, but also the quantitative one pertaining to the animal and landscape in

question, are important." The last part of the sentence seems like a non sequitor, as modelling provides quantitative patterns.

Page 17

Line 9. Change 'uncertainty in predictions were' to 'uncertainty in prediction was'

Page 18

Line 2. I am not sure what 'worming' means here. Do buzzards eat worms?

Therefore, on the one hand I think that some of the model output that I could understand is very nice. On the other hand, the study comes off as a 'methods' paper and is poorly written. At the very least the manuscript needs a thorough revision. Maybe there is more that the authors could do with the results to link the results to some more general landscape level principles. Otherwise, I would recommend that the authors, after thorough revision, submit the manuscript to a journal like "Methods in Ecology and Evolution" or "Ecological Modelling"

Referee: 2

Comments to the Author(s)

Anticipation of animal population patterns in a changing landscape

Eduardo M. Arraut, Sean W. Walls, David W. Macdonald and Robert E. Kenward

The paper illustrates well how data obtained with Resource-Area-Dependence Analysis (RADA), which quantifies the resource needs of individuals, can be used in Agent-based Modelling (ABM) to anticipate changes in animal populations in response to landscape alterations. The paper also illustrates how this approach can be used to predict how animal populations are likely to respond to various scenarios of landscape change.

The model species in the buzzard, a bird for which the authors have good data on the use of landscapes. The paper is generally well written, and the figures are adequate. However, I believe that there are a few issues that should be addressed to improve the paper.

The introduction is formatted somewhat like a "state of the art" and some key elements usually included in the introductions of papers are missing. Most obvious is the lack of the presentation of the objectives of the paper. These are apparently included only in the modelling subsection of the Methods, but readers usually look for them in the introduction. I also believe that it would be important to add to the introduction a paragraph explaining what is new in the paper. Why is it important to carry out this study? How does it differ from other modeling studies and approaches?

The discussion section includes a good text about the limitations of the modelling approach used in the study. However, I believe that it would gain by including a discussion on the relative merits of the proposed approach compared to other existing modelling approaches with similar objectives. Based on the results obtained in the study, can you recommend the approach followed? Are the results likely to be better than those that you could have obtained using other existing approaches? Why? Under what circumstances is the proposed one better?

A few specific comments:

P5L8 "In a previous study, we used remote sensing data collected over five years to apply Resource-Area-Dependency Analysis (RADA), using the buzzard tracking data and the Land Cover Map of Great Britain of 1990s."

Comment: Remote sensing is nowadays a very broad concept... I suggest that you indicate what type of remote sensing data you are referring to and what it was used for, in the modelling.

P8L6 Our general purpose was to test whether an ABM of a wide-ranging animal...

Comment: I suggest that you avoid using the word "test", because strictly speaking you did not really test this.

P12 L13 "2. Entities, state variables and scales"

Comment: I believe that most of the information in this section would be easier to consult if it were presented as a table or diagram.

Author's Response to Decision Letter for (RSPB-2020-2748.R0)

See Appendix A.

RSPB-2021-0993.R0

Review form: Reviewer 1

Recommendation

Accept with minor revision (please list in comments)

Scientific importance: Is the manuscript an original and important contribution to its field?

Excellent

General interest: Is the paper of sufficient general interest?

Excellent

Quality of the paper: Is the overall quality of the paper suitable?

Excellent

Is the length of the paper justified?

Yes

Should the paper be seen by a specialist statistical reviewer?

Yes

Do you have any concerns about statistical analyses in this paper? If so, please specify them explicitly in your report.

No

It is a condition of publication that authors make their supporting data, code and materials available - either as supplementary material or hosted in an external repository. Please rate, if applicable, the supporting data on the following criteria.

Is it accessible?

Yes

Is it clear?

Yes

Is it adequate?

Yes

Do you have any ethical concerns with this paper?

No

Comments to the Author

The authors use buzzard tracking data in an agent-based model (ABM) to predict the distribution and estimate the potential population of buzzards in a 22 km x 6 km area of southern Dorset. They also project the potential changes in population if areas of 30%, 60% and 90% of meadow are converted to woodland. To distribute territories in the simulated landscape, the authors use a minimum requirements hypothesis. That is, they assume that each individual buzzard, after landing at a randomly chosen woodlot, with attempt to add close free rough-ground and then meadow areas to its territory until it reaches its minimum needs. This is done with each new simulated buzzard until a saturation of territories is reached. The authors compare the simulation with empirical information. The model works remarkably well at describing current data on buzzards. The authors then use the model to project buzzard populations when 30%, 60%, and 90% of meadows are converted to woodland.

I reviewed an earlier version of this manuscript. The authors have greatly improved the manuscript. Importantly, the emphasis has shifted from the emphasis on methodology to predicting the abundance of the buzzard under projections of future land use changes; that is, the importance of understanding all the potential consequences of changes in landscape. I like the comment in Line 60 on considering 'the holistic cascade of effect of their interventions'. It is an excellent point, and the manuscript is a good illustration of considering such holistic effects, on one species in this case, but with obvious further effects. Secondly, the manuscript is much better written.

I have a few additional comments.

Line 83. Maybe more can be said about what the importance of 'perimeter' is. It is defined for inner and outer convex polygons (Lines 255, 299), but is there some biological significance for why it is one of the five dimensions. Is there an upper limit on what a perimeter length can be?

Line 88. What is a 'problem buzzard'? Preying on chickens?

Line 95. Concerning the 'Resource Dispersion Hypothesis', it would be useful to very briefly define it; as, e.g., 'the size and shape of territories depends on the dispersion of materials needed for survival and reproduction', which, as the authors note, is similar to RADA.

Line 122. LCMBG was already defined in Line 91.

Lines 233-239. Concerning the sensitivity analysis, the Morris screening method is described in the SI, but a short sentence here would be helpful.

Line 320. I don't think UKCEH (UK Centre for Ecology & Hydrology) is defined anywhere.

Lines 369-372. Sentence is awkward,

Lines 363-379. The authors here talk about Resource Selection Functions (RSF) as 'the most popular way to predict animal population patterns' in a real landscape, presumably to compare that approach with their ABM approach. But this paragraph is very hard to understand, especially for anyone not familiar with the RSF approach. I think such a comparison is perhaps useful, but it must be improved.

Over, this is an interesting and useful manuscript. The authors are able to show that a straightforward application of ABM, with simple assumptions that avoid the need to simulate bioenergetics and dynamic interactions of individuals, is able to predict the pattern of buzzard territories in a landscape to a sufficient degree of accuracy. This is important from the conservation standpoint for buzzards, as well as showing a methodology that could be applied elsewhere.

Decision letter (RSPB-2021-0993.R0)

07-May-2021

Dear Dr Moraes Arraut

I am pleased to inform you that your manuscript RSPB-2021-0993 entitled "Anticipation of common buzzard population patterns in the changing UK landscape" has been accepted for publication in Proceedings B.

The referee(s) have recommended publication, but also suggest some minor revisions to your manuscript. Therefore, I invite you to respond to the referee(s)' comments and revise your manuscript. Because the schedule for publication is very tight, it is a condition of publication that you submit the revised version of your manuscript within 7 days. If you do not think you will be able to meet this date please let us know.

[http://datadryad.org/submit?journalID=RSPB&manu=\(Document not available\)](http://datadryad.org/submit?journalID=RSPB&manu=(Document not available)) which will take you to your unique entry in the Dryad repository. If you have already submitted your data to dryad you can make any necessary revisions to your dataset by following the above link.

Please see <https://royalsociety.org/journals/ethics-policies/data-sharing-mining/> for more details.

Sincerely,

Dr Robert Barton

Reviewer(s)' Comments to Author:

Referee: 1

Comments to the Author(s).

The authors use buzzard tracking data in an agent-based model (ABM) to predict the distribution and estimate the potential population of buzzards in a 22 km x 6 km area of southern Dorset. They also project the potential changes in population if areas of 30%, 60% and 90% of meadow are converted to woodland. To distribute territories in the simulated landscape, the authors use a minimum requirements hypothesis. That is, they assume that each individual buzzard, after landing at a randomly chosen woodlot, with attempt to add close free rough-ground and then meadow areas to its territory until it reaches its minimum needs. This is done with each new simulated buzzard until a saturation of territories is reached. The authors compare the simulation with empirical information. The model works remarkably well at describing current data on buzzards. The authors then use the model to project buzzard populations when 30%, 60%, and 90% of meadows are converted to woodland.

I reviewed an earlier version of this manuscript. The authors have greatly improved the manuscript. Importantly, the emphasis has shifted from the emphasis on methodology to

predicting the abundance of the buzzard under projections of future land use changes; that is, the importance of understanding all the potential consequences of changes in landscape. I like the comment in Line 60 on considering 'the holistic cascade of effect of their interventions'. It is an excellent point, and the manuscript is a good illustration of considering such holistic effects, on one species in this case, but with obvious further effects. Secondly, the manuscript is much better written.

I have a few additional comments.

Line 83. Maybe more can be said about what the importance of 'perimeter' is. It is defined for inner and outer convex polygons (Lines 255, 299), but is there some biological significance for why it is one of the five dimensions. Is there an upper limit on what a perimeter length can be?

Line 88. What is a 'problem buzzard'? Preying on chickens?

Line 95. Concerning the 'Resource Dispersion Hypothesis', it would be useful to very briefly define it; as, e.g., 'the size and shape of territories depends on the dispersion of materials needed for survival and reproduction', which, as the authors note, is similar to RADA.

Line 122. LCMBG was already defined in Line 91.

Lines 233-239. Concerning the sensitivity analysis, the Morris screening method is described in the SI, but a short sentence here would be helpful.

Line 320. I don't think UKCEH (UK Centre for Ecology & Hydrology) is defined anywhere.

Lines 369-372. Sentence is awkward,

Lines 363-379. The authors here talk about Resource Selection Functions (RSF) as 'the most popular way to predict animal population patterns' in a real landscape, presumably to compare that approach with their ABM approach. But this paragraph is very hard to understand, especially for anyone not familiar with the RSF approach. I think such a comparison is perhaps useful, but it must be improved.

Over, this is an interesting and useful manuscript. The authors are able to show that a straightforward application of ABM, with simple assumptions that avoid the need to simulate bioenergetics and dynamic interactions of individuals, is able to predict the pattern of buzzard territories in a landscape to a sufficient degree of accuracy. This is important from the conservation standpoint for buzzards, as well as showing a methodology that could be applied elsewhere.

Author's Response to Decision Letter for (RSPB-2021-0993.R0)

See Appendix B.

Decision letter (RSPB-2021-0993.R1)

14-May-2021

Dear Dr Arraut

I am pleased to inform you that your manuscript entitled "Anticipation of common buzzard population patterns in the changing UK landscape" has been accepted for publication in Proceedings B.

Data Accessibility section

Open Access

Paper charges

Sincerely,

Proceedings B

Appendix A

16-Dec-2020

Dear Dr Moraes Arraut:

I am writing to inform you that your manuscript RSPB-2020-2748 entitled "Anticipation of animal population patterns in a changing landscape" has, in its current form, been rejected for publication in Proceedings B.

This action has been taken on the advice of referees, who have recommended that substantial revisions are necessary. With this in mind we would be happy to consider a resubmission, provided the comments of the referees are fully addressed. However please note that this is not a provisional acceptance.

Sincerely,

Dr Robert Barton
mailto:proceedingsb@royalsociety.org

Thank you very much for giving us the chance to improve our manuscript by taking into consideration the very important points raised by the AE and referees. We believe we have addressed every single point they raised. After each AE or referee comment, we have included our response (preceded by '##'), the line number of the tracked changes document and, immediately after it and in brackets, the line number of the clean document; e.g. L. 182(162) is line number 182 of the tracked changes document and line number 162 of the clean document.

Associate Editor
Board Member: 1
Comments to Author:

I looked at the ms and the reviews. I like about this ms is the integration of data and modelling and making it work. This is interesting and important. However, the ms would be much more far reaching if the results address wider ecological issues or questions. The reviewers make the same points. Addressing these points will require changes to introduction and discussion. At the moment, the paper has an interesting method but as it is I do not think that is of sufficient interest for the readership of this journal. To make it suitable for a wide biological audience the focus should come away from the just the method and include wider ecological

aspects. The paper would benefit if additional hypotheses would be tested, or if it could explore wider landscape ecology aspects.

Thank you very much for these very important points, which we believe have helped us improve our manuscript considerably. After profound changes to Introduction and Discussion, the main focus of the manuscript is now on the wider landscape ecology aspect of anticipating how the on-going changes to the UK landscape, motivated by other priorities - flood management, changing human diets and climate change mitigation -, will likely affect the common buzzard, a protected species with a long history of negative impacts from humans in the UK.

We note that during revision we detected a small error in the table presenting the final calibration parameters combination, which had passed unnoticed by our cross-checking because it affected only, and to a very small extent, the less important of the four model parameters. We corrected this small error and re-ran all analyses steps posterior to calibration (hypothesis testing concerning defence of key resource patches, local and global sensitivity analyses, and final results), and then produced the updated figures and tables in the main manuscript and SI. The effect on final results was negligible (~1% increase in the prediction for final abundance in the 1990 UK landscape, and no appreciable differences to the other outputs), so no changes to the interpretation of the results in the Discussion were needed.

Reviewer(s)' Comments to Author:

Referee: 1

Comments to the Author(s)

The authors use buzzard tracking data in an agent-based model to predict the distribution and estimate the potential population of buzzards in a 22 km x 6 km area of southern Dorset. They also project the potential changes in population if areas of 30%, 60% and 90% of meadow are converted to woodland.

The model developed here seems to be properly formulated and implemented. The comparison with empirical data is quite good. Predicted abundance is 9% greater than field-based estimates. Such a small difference could easily be explained by a number of factors, so it and other comparisons seem to validate the model. Figure 4 is very nice.

One problem that I have with the manuscript is that it seems to be a 'methods paper' rather than addressing an ecological issue. It is true that the results seem to confirm hypotheses on the habitat needs of the buzzards. But the manuscript seems mostly aimed at showing how well the methodology works. The 30%, 60% and 90% hypothetical conversions of meadow to woodland appear to be projected merely for providing something to apply the model to. I think that a lot more could be learned from this exercise by using the results so far, or some extensions, to explore aspects of landscape ecology theory, such as a more thorough study of how various landscape factors, including configuration of habitat types, affect consumer abundance.

Thank you very much for this very important point. In line with one of the options given to us by the AE, after profound changes to Introduction and Discussion the manuscript now focuses on the wider landscape ecology issue of anticipating how the on-going changes to the UK landscape will likely affect the common buzzard, as noted above for the Associate Editor.

A second problem is that the manuscript is very poorly written including a failure to clearly define some of main parameters and assumptions.

Thank you for this important comment. We have made great efforts to improve the text and have included the parameters table, Table 1, L. 865(624), and stated the assumptions more clearly at L. 287(194).

I have detailed comments below. Major and minor comments are mixed. Page numbers used in this review refer to the numbers at the bottom of the page.

Page 3

Abstract, Line 6. I think change 'when animal and fieldsite were small' to 'when animals and fieldsites were small', although I am not sure.

Thank you. This sentence does not appear in the new introduction.

Page 5

Line 11. 'three degrees of realisation'. I am not sure what those are as I didn't see reference to that term later. I am guessing they are size, shape and pattern of overlap, but I am not sure.

The 'three degrees of realisation' referred to the realisations of the landscape change scenarios consisting of 30%, 60% and 90% of meadow patches >20 ha being converted into woodland. In the revised manuscript we refer to each of these different extents of meadow conversion as an independent scenario (when we mention the possibility that 'other realistic landscape change scenarios, could be explored...'), L. 141(97).

Line 20. Delete comma after 'UK'

Sentence removed from new Introduction.

Page 6.

Somewhere around here, or perhaps earlier, the authors need to provide a paragraph or so on buzzard ecology; at least enough to have a succinct explanation of how they use habitat types. As it is, the reader has to piece together bits of information scattered through the manuscript.

Thank you. We have included a paragraph about buzzard conservation status and resource use in the Introduction, L. 101(63), and kept the explanation about other aspects of its spatial ecology in the Methods, L. 174(104).

Page 7

Lines 10-11 "in which meadow is gradually substituted for wood". I think the opposite is meant, "in which wood is gradually substituted for meadow".

The landscape change scenarios represent the on-going trend in meadow being replaced by woodland in the UK landscape. This is made clear at L. 90-94(52-56) and 208(135), and in terms of impact on buzzards at L. 112(74).

Line 16. 'rough ground'. I could find no place where that is defined.

Thank you for this important point, now addressed in L. 111(73).

Page 9

Line 4. Change 'from where' to 'from which'

Done, L. 255(164).

Page 10

Lines 16-18 "we assessed the utilization distributions as outlines at 5% intervals, from 30% to 80%, using 80% as proxy for the smallest core encompassing all resources because buzzard behaviour involves more flight activity in the outermost 15% of locations." I don't understand this sentence. It needs to be revised.

Thank you for pointing this out. We have revised this sentence, L. 299(205).

Page 12.

"to assess the relative importance of rough-ground area and distance and meadow area and distance". I am not sure what is meant by 'distance' here. The word 'distance' is used in various ways in the manuscript and it is often not clear what it means in particular cases.

Thank you for this important point. We meant 'forage search distance', as reflecting an energetic constraint upon how far the buzzard can fly to search for resources. These two forage search distance parameters, for rough-ground and for meadow, and all other parameters in the model, are now defined in Table 1, L. 865(624). Small changes to Table 2, L. 871(628), and the '3. Process overview and scheduling' section, L. 246(155) were also made to ensure consistency.

Page 14

Line 2. Insert comma after 'neighbors'

Thank you. Done at L. 380(276).

Line 17. 'spreading out of meadows'. Does that mean that the density of patches of meadows is decreasing?

Exactly. In the landscape change scenarios representing 30%, 60% and 90% conversion of meadow fields > 20 ha into woodland, the density of meadow patches decreases progressively. For increased clarity, we have changed this to '...reduction and dispersion of meadow, ...', L. 394(289).

Page 15

Lines 2-3. "LSA showed that abundance was inversely regulated mainly by the individual's meadow area requirement." I am not sure what the term 'inversely regulated' means and I don't really understand Figures S9 and S10. I don't know what Morris Screening is in Figure S9 - it should be explained. In Figure S10, which I magnified to the maximum possible on my laptop, but could still only read 'MCP' on the y-axis, and I don't know what that means - some sort of perimeter perhaps. The figure captions here are inadequate.

Thank you for these important points. We have changed the sentence about LSA, L. 329(233). We have also included, in the SI, an explanation about the Morris Screening Method, starting at L. 714, plus a sentence in the caption of Fig.S9, L. 750, with a succinct interpretation of the graphs, and we now also state in the main document, L. 334(238), that the modified Morris screening method is explained in the SI. We have also changed the y-axis titles in Fig. S10, L. 760, – and yes, you are right, MCP is Minimum Convex Polygon outline -, and included in its caption a succinct interpretation of its panels, L. 765.

Lines 15-17. "Modelling alone seems insufficient, because in applied cases not only the qualitative population patterns, but also the quantitative one pertaining to the animal and landscape in question, are important." The last part of the sentence seems like a non sequitor, as modelling provides quantitative patterns.

Thank you for pointing this out. This sentence was removed from the new Discussion.

Page 17

Line 9. Change 'uncertainty in predictions were' to 'uncertainty in prediction was'

Done, L 460(342).

Page 18

Line 2. I am not sure what 'worming' means here. Do buzzards eat worms?

Yes. During winter, when small mammals are very difficult to catch, buzzards eat invertebrates, mainly worms. For clarity, we have changed this sentence to "...may deter buzzards from foraging for worms in meadow...", L. 477(358).

Therefore, on the one hand I think that some of the model output that I could understand is very nice. On the other hand, the study comes off as a 'methods' paper and is poorly written. At the very least the manuscript needs a thorough revision. Maybe there is more that the authors could do with the results to link the results to some more general landscape level principles. Otherwise, I would recommend that the authors, after thorough revision, submit the manuscript to a journal like "Methods in Ecology and Evolution" or "Ecological Modelling"

We are very grateful to you for having devoted time to help us improve our manuscript.

Referee: 2

Comments to the Author(s)

Anticipation of animal population patterns in a changing landscape

Eduardo M. Arraut, Sean W. Walls, David W. Macdonald and Robert E. Kenward

The paper illustrates well how data obtained with Resource-Area-Dependence Analysis (RADA), which quantifies the resource needs of individuals, can be used in Agent-based Modelling (ABM) to anticipate changes in animal populations in response to landscape alterations. The paper also illustrates how this approach can be used to predict how animal populations are likely to respond to various scenarios of landscape change.

The model species in the buzzard, a bird for which the authors have good data on the use of landscapes. The paper is generally well written, and the figures are adequate. However, I believe that there are a few issues that should be addressed to improve the paper.

The introduction is formatted somewhat like a "state of the art" and some key elements usually included in the introductions of papers are missing. Most obvious is the lack of the presentation of the objectives of the paper. These are apparently included only in the modelling subsection of the Methods, but readers usually look for them in the introduction.

Thank you for this fundamental point. The objective is now clearly stated in the Introduction, L. 114(76).

I also believe that it would be important to add to the introduction a paragraph explaining what is new in the paper. Why is it important to carry out this study? How does it differ from other modeling studies and approaches?

Thank you for these helpful recommendations. We have added a paragraph that explains why this study was important and how the modelling approach used here differs from other approaches, starting at L. 122(78), and expanded on the advantage over other modelling in the Discussion at L# (see below).

The discussion section includes a good text about the limitations of the modelling approach used in the study. However, I believe that it would gain by including a discussion on the relative merits of the proposed approach compared to other existing modelling approaches with similar objectives. Based on the results obtained in the study, can you recommend the approach followed? Are the results likely to be better than those that you could have obtained using other existing approaches? Why? Under what circumstances is the proposed one better?

To explain the points raised here, we have included a new section in the Discussion, starting at L. 491(362). Specifically, we explain how, for predicting animal distribution and abundance in changing landscapes, our combined approach (RADA-ABM) improves on Resource Selection Functions, starting at L. 497(363), and on ABM alone, starting at L. 529(381). We also elaborate further on the particular strengths of the approach we used, starting at L. 542(394).

A few specific comments:

P5L8 "In a previous study, we used remote sensing data collected over five years to apply Resource-Area-Dependency Analysis (RADA), using the buzzard tracking data and the Land Cover Map of Great Britain of 1990s."

Comment: Remote sensing is nowadays a very broad concept... I suggest that you indicate what type of remote sensing data you are referring to and what it was used for, in the modelling.

Thank you. We used radio-tracking and categorical satellite mapping, and have now specified this in L. 543(395).

P8L6 Our general purpose was to test whether an ABM of a wide-ranging animal...

Comment: I suggest that you avoid using the word "test", because strictly speaking you did not really test this.

Thank you very much for pointing this out. You are right. The *Purpose* section was re-written, L. 205(132), in light of the greater focus of the manuscript on buzzards and landscape change, following suggestions by R1 and the AE.

P12 L13 "2. Entities, state variables and scales"

Comment: I believe that most of the information in this section would be easier to consult if it were presented as a table or diagram.

Excellent point, thank you. Entities and state variables are now presented in Table 1, L. 865(624).

We are very grateful to you for having devoted time to help us improve our manuscript.

Appendix B

07-May-2021

Dear Dr Moraes Arraut

I am pleased to inform you that your manuscript RSPB-2021-0993 entitled "Anticipation of common buzzard population patterns in the changing UK landscape" has been accepted for publication in Proceedings B.

The referee(s) have recommended publication, but also suggest some minor revisions to your manuscript. Therefore, I invite you to respond to the referee(s)' comments and revise your manuscript. Because the schedule for publication is very tight, it is a condition of publication that you submit the revised version of your manuscript within 7 days. If you do not think you will be able to meet this date please let us know.

If you wish to submit your data to Dryad (<http://datadryad.org/>) and have not already done so you can submit your data via this link [http://datadryad.org/submit?journalID=RSPB&manu=\(Document not available\)](http://datadryad.org/submit?journalID=RSPB&manu=(Document not available)) which will take you to your unique entry in the Dryad repository. If you have already submitted your data to dryad you can make any necessary revisions to your dataset by following the above link.

Please see <https://royalsociety.org/journals/ethics-policies/data-sharing-mining/> for more details.

Sincerely,

Dr Robert Barton

Thank you very much again for giving us the chance to improve our manuscript by taking into consideration the very important points raised by the referee. We believe we have addressed every single point. As in the previous submission, after each referee comment we have included our response (preceded by '##'), the line number of the tracked changes document and, immediately after it and in brackets, the line number of the clean document; e.g. L. 182(162) is line number 182 of the tracked changes document and line number 162 of the clean document.

Reviewer(s)' Comments to Author:

Referee: 1

Comments to the Author(s).

The authors use buzzard tracking data in an agent-based model (ABM) to predict the distribution and estimate the potential population of buzzards in a 22 km x 6 km area of southern Dorset. They

also project the potential changes in population if areas of 30%, 60% and 90% of meadow are converted to woodland. To distribute territories in the simulated landscape, the authors use a minimum requirements hypothesis. That is, they assume that each individual buzzard, after landing at a randomly chosen woodlot, with attempt to add close free rough-ground and then meadow areas to its territory until it reaches its minimum needs. This is done with each new simulated buzzard until a saturation of territories is reached. The authors compare the simulation with empirical information. The model works remarkably well at describing current data on buzzards. The authors then use the model to project buzzard populations when 30%, 60%, and 90% of meadows are converted to woodland.

I reviewed an earlier version of this manuscript. The authors have greatly improved the manuscript. Importantly, the emphasis has shifted from the emphasis on methodology to predicting the abundance of the buzzard under projections of future land use changes; that is, the importance of understanding all the potential consequences of changes in landscape. I like the comment in Line 60 on considering 'the holistic cascade of effect of their interventions'. It is an excellent point, and the manuscript is a good illustration of considering such holistic effects, on one species in this case, but with obvious further effects. Secondly, the manuscript is much better written.

I have a few additional comments.

Line 83. Maybe more can be said about what the importance of 'perimeter' is. It is defined for inner and outer convex polygons (Lines 255, 299), but is there some biological significance for why it is one of the five dimensions. Is there an upper limit on what a perimeter length can be?

Thank you for raising this point. Mathematically, the perimeter length of minimum convex polygons (MCP) with the same area but different shapes could tend to infinity (by having MCPs that are progressively narrower and more elongated). Naturally, however, the perimeter length of an MCP for a wild buzzard is limited, by the individual's energetic budget and the grain size of resource patches. For a virtual buzzard in our model, the maximum possible perimeter length would be 11.35 km, and would result from an (unrealistic) linear territory in which all the required resource patches were positioned side by side (1 woodland + 9 rough-ground + 216 meadow 25x25 m patches). In the manuscript, we have modified the text as follows at L. 297-301(84-88): "Model predictions were assessed by comparison with knowledge of wild buzzards according to five dimensions of their spatial ecology, namely home-range area, perimeter, pairwise overlaps and population distribution and abundance. Since two home-ranges can have the same areas and yet very different perimeters, and vice-versa, home-range area and perimeter are two proxies for energy expenditure from movement to acquire resources which could differ according to buzzard foraging behaviour. Pairwise home-range overlaps are a proxy for territorial spacing."

Line 88. What is a 'problem buzzard'? Preying on chickens?

The main problem arises from buzzards preying on poultry or game birds. We have modified at L. 284(71) – where the concept of 'problem buzzard' is first mentioned: "Nowadays, general attitudes towards buzzards in the UK are more positive, with only occasional, local episodes of human-wildlife conflict attributed to 'problem individuals', which prey on poultry or game birds."

Line 95. Concerning the 'Resource Dispersion Hypothesis', it would be useful to very briefly define it; as, e.g., 'the size and shape of territories depends on the dispersion of materials needed for survival and reproduction', which, as the authors note, is similar to RADA.

Good point, thank you. We have modified as follows at L. 312-315(99-101): "The RADA process has conceptual parallels to the Resource Dispersion Hypothesis, which postulates that territory size depends on the dispersion of the resources needed for survival and reproduction,

and which has been used to explain the territories of >40 species in five continents [20].”

Line 122. LCMBG was already defined in Line 91.

Well spotted, thank you. Now erased from L. 341(127).

Lines 233-239. Concerning the sensitivity analysis, the Morris screening method is described in the SI, but a short sentence here would be helpful.

Good point, thank you. Now modified at L. 454-456(239-242): “LSA used a modified version of the Morris screening method, which makes no assumptions about the model and uses individually randomised one-factor-at-a-time designs to assess the effects of changes in parameter values on outputs [33,34]. The modified Morris screening was used to assess the relative importance...”.

Line 320. I don't think UKCEH (UK Centre for Ecology & Hydrology) is defined anywhere.

Well spotted, thank you. Now defined at L. 542(328).

Lines 369-372. Sentence is awkward,

Thank you. This sentence does not appear in the updated paragraph (see below).

Lines 363-379. The authors here talk about Resource Selection Functions (RSF) as 'the most popular way to predict animal population patterns' in a real landscape, presumably to compare that approach with their ABM approach. But this paragraph is very hard to understand, especially for anyone not familiar with the RSF approach. I think such a comparison is perhaps useful, but it must be improved.

Excellent point, thank you. We have modified the entire paragraph starting at L. 585(371): “Resource Selection Functions (RSF) seem to be the most popular method to predict animal distribution and abundance in the geographical space of a real landscape. A RSF can predict these population patterns from correlations between a wide variety of data about animal presence, e.g. spoor or GPS tracking, and for resources or conditions, e.g. land-cover or altitude [46,47]. Important assumptions of a RSF are that 1) the animal population is at equilibrium density or following an ideal free distribution when the calibration data are collected, 2) abundance does not depend on factors other than resources, and 3) the availabilities of the resources in the calibration and extrapolation landscapes is similar [46]. An example of when assumptions (1-2) would not have been reasonable was shown with an ABM for oystercatchers in the Exe estuary, UK. Mortality was found to be influenced by interference competition only after a certain density threshold, so a RSF built using data collected when density was below this threshold would overestimate maximum abundance by a considerable margin [43]. Additionally, when the resources are found within a landscape category that is being impacted by human action, e.g. meadow for buzzards being replaced by woodland, assumption (3) may be hard to meet and therefore extrapolations to future landscapes may be problematic. Thus, the assumptions underlying a RSF can restrict applicability to certain situations that may be particularly important for conservation, such as when an animal population is below equilibrium density owing to endangerment or recurrent perturbations, or when the aim is to assess the impact of landscape change on wildlife.”

Over, this is an interesting and useful manuscript. The authors are able to show that a straightforward application of ABM, with simple assumptions that avoid the need to simulate bioenergetics and dynamic interactions of individuals, is able to predict the pattern of buzzard territories in a landscape to a sufficient degree of accuracy. This is important from the conservation standpoint for buzzards, as well as showing a methodology that could be applied elsewhere.

We are very glad you liked our manuscript, and are very grateful to you for your precise and constructive criticisms in this and the first revision.